# Hybrid Quantum Technologies for Quantum Support Vector Machines

**Filippo Orazi *** , **Simone Gasperini** , **Stefano Lodi** and **Claudio Sartori**

Dipartimento di Informatica, Alma Mater Studiorum—University of Bologna, 40126 Bologna, Italy; simone.gasperini4@unibo.it (S.G.); stefano.lodi@unibo.it (S.L.); claudio.sartori@unibo.it (C.S.)
* Correspondence: filippo.orazi2@unibo.it

**Abstract:** Quantum computing has rapidly gained prominence for its unprecedented computational efficiency in solving specific problems when compared to classical computing counterparts. This surge in attention is particularly pronounced in the realm of quantum machine learning (QML) following a classical trend. Here we start with a comprehensive overview of the current state-of-the-art in Quantum Support Vector Machines (QSVMs). Subsequently, we analyze the limitations inherent in both annealing and gate-based techniques. To address these identified weaknesses, we propose a novel hybrid methodology that integrates aspects of both techniques, thereby mitigating several individual drawbacks while keeping the advantages. We provide a detailed presentation of the two components of our hybrid models, accompanied by the presentation of experimental results that corroborate the efficacy of the proposed architecture. These results pave the way for a more integrated paradigm in quantum machine learning and quantum computing at large, transcending traditional compartmentalization.

**Keywords:** quantum computing; quantum machine learning; quantum support vector machine; quantum annealling; gate-based quantum computation

## 1. Introduction

Quantum computing, as an evolving sub-discipline within computer science, amalgamates various research fields, including physics, mathematics, and computer science, among many others. The course of technological advancement has divided the field into two distinct approaches: gate-based quantum computation and adiabatic quantum computation. Although these approaches are theoretically equivalent, their practical utilization diverges significantly. Gate-based quantum computation is studied for a more general-purpose application, whereas adiabatic computation is primarily employed for quantum annealing, an optimization process aimed at determining the minimum of a cost function.

Quantum machine learning (QML) stands as a heavily studied sub-discipline within the framework of quantum computation, it being popular within both the gate-based and adiabatic paradigms. This article focuses on the Support Vector Machine (SVM) model and describes how these two divergent quantum computing approaches seek to implement it. We propose a novel approach to model this problem, combining both technologies.

Our model is based on the observation that the two quantum approaches for the Quantum Support Vector Machine (QSVM) focus on two different parts of the classical approach. Gate-based computation aims to leverage quantum properties to discover a useful kernel for the high-dimensional Hilbert space. On the other hand, the annealing model focuses on optimization that comes after computing the kernel matrix. These are two separate but complementary components of the Support Vector Machine model. The combination of these approaches, if correctly applied, provides advantages over the classical method, as well as each approach on its own.

The following sections perform an in-depth analysis of the current advancements in Quantum Support Vector Machines. It begins with an exposition on classical SVM, then delves into an investigation of the distinct approaches employed in computing QSVM.

### 1.1. Classical Support Vector Machines

An SVM is a supervised machine learning algorithm designed for both classification and regression tasks. It operates on a dataset $D\{(x_n, t_n) : n = 0, \ldots, N - 1\}$, where $x_n \in \mathbb{R}^d$ represents a point in $d$-dimensional space, serving as a feature vector, and $t_n$ denotes the target label assigned to $x_n$. We will focus on the classification task and learning a binary classifier that assigns a class label $\hat{t}_n = \pm 1$ to a given data point $x_n$. For clarity, we designate the class $t_n = 1$ as 'positive' and the class $t_n = -1$ as 'negative'.

The training of an SVM entails solving the quadratic programming (QP) problem:

$$minimize \ \ E = \frac{1}{2} \sum_{nm} \alpha_n \alpha_m t_n t_m k(x_n, x_m) - \sum_n \alpha_n \tag{1}$$

with

$$0 \leq \alpha_n \leq C \quad and \quad \sum_n \alpha_n t_n = 0 \tag{2}$$

For a set of $N$ coefficients $\alpha_n \in \mathbb{R}$, where $C$ denotes a regularization parameter and $k(\cdot, \cdot)$ represents the kernel function of the Support Vector Machine [1,2], the resulting coefficients $\alpha_n$ establish a $(d - 1)$-dimensional decision boundary that partitions $\mathbb{R}^d$ into two regions corresponding to the predicted class label. The decision boundary is defined by points associated with $\alpha_n \neq 0$, commonly referred to as the support vectors of the SVM. Prediction for an arbitrary point $x \in \mathbb{R}^d$ can be accomplished by

$$f(x) = \sum_n \alpha_n t_n k(x_n, x) + b \tag{3}$$

where $b$ can be estimated by formula [2]:

$$b = \frac{\sum_n \alpha_n (C - \alpha_n)[t_n - \sum_m \alpha_m t_m k(x_m, x_n)]}{\sum_n \alpha_n (C - \alpha_n)} \tag{4}$$

Geometrically, the decision function $f(x)$ corresponds to the signed distance between the point $x$ and the decision boundary. Consequently, the predicted class label $\hat{t}$ for $x$ as determined by the trained Support Vector Machine is given by $\hat{t} = \text{sign}(f(x))$.

The problem formulation can be equivalently expressed as a convex quadratic optimization problem [3], indicating its classification among the rare minimization problems in machine learning that possess a globally optimal solution. It is crucial to note that while the optimal solution exists, it is dataset-specific and may not necessarily generalize optimally across the entire data distribution.

Kernel-based SVMs exhibit exceptional versatility as they can obtain nonlinear decision boundaries denoted by $f(x) = 0$. This is achieved through the implicit mapping of feature vectors into higher-dimensional spaces [4]. Importantly, the computational complexity does not escalate with this higher dimensionality, as only the values of the kernel function $k(x_n, x_m)$ are involved in the problem specification. This widely recognized technique is commonly referred to as the "kernel trick" and has been extensively explored in the literature [1,2].

The selection of the kernel function significantly influences the outcomes, with radial basis function kernels (RBF) [3] generally serving as the search starting point for the right kernel of a Support Vector Machine problem. An RBF kernel is distinguished by the property that $k(x_n, x_m)$ can be expressed as a function of the distance $\|x_n - x_m\|$ [1].

The Gaussian kernel, often referred as "The RBF kernel", is the most prevalent RBF kernel and is represented as

$$rbf(x_n, x_m) = e^{-\eta||x_n - x_m||^2} \qquad (5)$$

Here, the value of the hyperparameter $\eta > 0$ is typically determined through a calibration procedure before the training phase [5].

The SVM are very susceptible to the choice of the hyper-parameters, like $\eta$ or $C$, and different assignments can radically change the result of the optimization.

### 1.2. Annealer-Based Quantum Support Vector Machines

Quantum computers leverage diverse hardware approaches and technologies, with quantum annealing being one notable paradigm.

Quantum annealing commences by establishing a quantum-mechanical superposition of all possible states, followed by the system's evolution governed by the time-dependent Schrödinger equation. The amplitudes of all states undergo continuous changes and, if the rate of change is sufficiently slow, the system remains in the ground state of the instantaneous Hamiltonian, characterizing adiabatic quantum computation [6]. In the case of insufficiently slow changes, the system may leave the ground state temporarily, while at the same time producing a higher likelihood of concluding in the ground state of the final problem Hamiltonian in a diabatic quantum computation [7,8]. Quantum annealers efficently solve problems formulated in either Quadratic Unconstrained Binary Optimization (QUBO) or Ising formulations.

Since we know how to formulate Support Vector Machines as convex quadratic optimization problems, a simple transformation is enough to achieve a QUBO formulation suitable for quantum annealing.

In a study by Willsch et al. [9], the authors introduced and explored the application of kernel-based Support Vector Machines on a DW2000Q quantum annealer [10]. From here on out, we refer to this methodology as Quantum Annealer Support Vector Machines (QaSVM). This approach offers distinct mathematical advantages, generating a spectrum of different classifiers with each iteration of the training process.

The study's findings demonstrate that the ensemble of classifiers produced by the quantum annealer can surpass the single classifier derived from classical SVMs when addressing the same computational problem. Performance is assessed through metrics such as Area Under the Receiver Operating Characteristic curve (AUROC), Area Under the Precision–Recall curve (AUPRC) [11,12], and accuracy. This advantage is attributed to the quantum annealer's ability to yield not only the global optimum for the training dataset but also a distribution of solutions close to optimality for the given optimization problem. The potential to combine these solutions enhances generalization to the test dataset.

Additional studies [13,14] corroborate the efficacy of QaSVMs and extend their promise to diverse problems, including multiclass classification.

### 1.3. Gate-Based Quantum Support Vector Machines

Gate-based quantum computers operate within the quantum circuit model, where quantum computation involves initialization of qubits to known values, the application of a sequence of quantum gates, and measurements. While this approach is broader in scope than the one outlined in Section 1.2, it presents formidable challenges in both hardware and algorithm development.

Quantum machine learning is predominantly studied on gate-based quantum computers, meaning there is more research on the QSVM model. THe quantum machine learning (QML) models exhibit diverse architectures and ansatz configurations for processing data. Even so, most models share a common initial step wherein classical data undergoes encoding in the Hilbert space, known as feature mapping. Schuld and Killoran [15] showed the equivalence of this phase to a quantum kernel and proposed one of the initial gate-based Support Vector Machine models (QgSVM). Their work outlines two approaches: firstly, they introduce the quantum kernel as a means to process classical data, transforming it

into new data with different dimensionality analogous to a classical kernel, which is then utilized in classical algorithms such as SVM. Secondly, they propose a parametric circuit for classifying input in a quantum environment. While this feature map–ansatz approach has become standard for Quantum Neural Networks (QNNs), a detailed discussion exceeds the scope of this document.

Research by Maria Schuld [16] aggregates results from various sources and concludes that all supervised quantum machine learning models (excluding generative ones) operate as kernel methods.

Another noteworthy contribution to the state of the art in gate-based QSVMs is the work by Havlicek et al. [17]. They introduce two SVM classifiers optimizing classically over data obtained using a quantum kernel trick to achieve quantum advantage. Similar to previous approaches, this involves non-linearly mapping data to a quantum state:

$$\phi : \tilde{x} \in \Omega \rightarrow |\phi(\tilde{x})\rangle\langle\phi(\tilde{x})| \tag{6}$$

The authors also highlight a crucial requirement for applying QgSVM: if the feature vector kernel $K(\tilde{x}, \tilde{z}) = |\langle\Phi(\tilde{x})|\Phi(\tilde{z})\rangle|^2$ is overly simplistic, such as generating only product states, it can be easily computed classically and loses its benefit. The advantage lies in leveraging the high dimensionality of the Hilbert space, necessitating a kernel that is impossible to simulate to surpass classical approaches.

## 2. Materials and Methods

In the outlined quantum computing landscape, we introduced two distinct architectures for employing Support Vector Machines in classification tasks. Both models, still in their early stages, warrant further thorough investigation and testing.

Quantum Annealer Support Vector Machines (QaSVM) demonstrate partial efficacy in addressing the problem, yet encounter challenges in precisely determining the optimal hyperplane. Additionally, QaSVM relies on a classical kernel trick to compute the $k(x_n, x_m)$ component of the formulation.

Conversely, Quantum gate-based Support Vector Machines (QgSVM) exhibit impressive data manipulation capabilities. However, they currently lack the readiness to handle the optimization aspect of the SVM algorithm. Present-day quantum technology does not possess the computational power necessary for intricate optimization problems. Consequently, the optimization step necessitates an approximate approach, achieved either through an ansatz (transitioning from QgSVM to a Quantum Neural Network, QNN model) or by resorting to classical optimization methods.

Our proposed approach involves a fusion of the quantum annealing and gate-based models, establishing a connection through a classical channel. The core idea is to capitalize on the strengths of each approach: annealing excels in optimization but lacks a dedicated kernel method, while the gate-based model performs well with the kernel trick but faces challenges in optimization.

Building upon our previous discussions, we recognize the ability to derive the Kernel Matrix from a quantum feature map within the gate-based architecture. Once the Kernel matrix is defined, we employ a standard procedure to reformulate the problem into a Quadratic Unconstrained Binary Optimization (QUBO) format, achieved by discretizing the continuous variables of a quantum computing problem. Subsequently, the QUBO formulation is encoded into the annealer, enabling the extraction of minima, as previously described. The illustrated process, as depicted in Figure 1, allows the attainment of results for even challenging problems, leveraging the combined advantages of both quantum technologies.

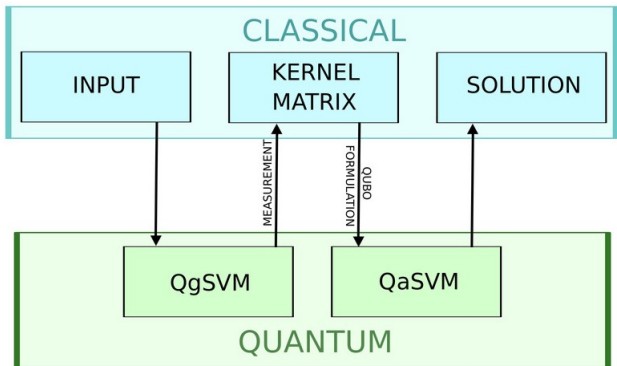

**Figure 1.** The image represents a detailed schematic outlining our proposed methodology for computing the Quantum Support Vector Machine (QSVM). The process initiates with the encoding of input data into the Quantum gate-based Support Vector Machine (QgSVM), which is responsible for the computation of the Kernel matrix. Following this initial stage, the Quantum annealer Support Vector Machine (QaSVM) encodes the problem's Quadratic Unconstrained Binary Optimization (QUBO) formulation. The subsequent solution extraction occurs through the quantum annealing process within the QaSVM framework. In the image the arrow show the passages from classical to quantum and viceversa. QgSVM and QaSVM are connected through the classical Kernel matrix by the measurement operation and the QUBO formulation.

### 2.1. From Classical SVM to QaSVM

We proceed to illustrate the translation of the Support Vector Machine into a form suitable for solving with Quantum Annealers. The complete formulation is detailed in the work by Willsch et al. [9].

The initial challenge in this transformation arises from the fact that, by definition, $\alpha_n \in \mathbb{R}$ while quantum annealers are only capable of producing discrete binary values. A straightforward resolution to this challenge involves binarizing the values of $\alpha$ as follows:

$$\alpha = \sum_{k}^{K-1} B^k a_{Kn+k} \tag{7}$$

where $a_{Kn+k} \in \{0,1\}$, $B$ represents the chosen basis and $K$ denotes the number of variables $a_k$ used for representing $\alpha$. We proceed by substituting $\alpha$ with $a$ and utilizing Equation (7) as a constraint, multiplied by a factor $\xi$:

$$E = \frac{1}{2} \sum_{nmkj} a_{Kn+k} a_{Km+j} B^{k+j} t_n t_m k(x_n, x_m)$$

$$- \sum_{nk} B^k a_{Kn+k} + \xi \left( \sum_{nk} B^k a_{Kn+k} t_n \right)^2 \tag{8}$$

$$= \sum_{n,m=0}^{N-1} \sum_{k,j=0}^{K-1} a_{Kn+k} \tilde{Q}_{Kn+k,Km+j} a_{Km+j} \tag{9}$$

where $\tilde{Q}$ is a matrix of $KN \times KN$ given by

$$\tilde{Q}_{Kn+k,Km+j} = \frac{1}{2} B^{k+j} t_n t_m (k(x_n, x_m) + \xi) - \delta_{nm} \delta_{kj} B^k \tag{10}$$

Naturally, given that $\tilde{Q}$ is symmetric, the upper triangular Quadratic Unconstrained Binary Optimization (QUBO) matrix $Q$ we seek is defined as $Q_{ij} = \tilde{Q}_{ij} + \tilde{Q}_{ji}$ for $i < j$, and $Q_{ii} = \tilde{Q}_{ii}$.

The last operation concludes the problem formulation so that it becomes suitable for a quantum annealer.

We do not delve into the embedding of the problem into the current available quantum annealing system for two primary reasons: Firstly, embedding is inherently tied to the architecture and could evolve with technological advancements, while the QUBO formulation remains constant. Secondly, in our experiments, we entrusted the embedding to the library's methods since they are optimized for this operation [18].

A pivotal aspect in formulating our new model involves recognizing that in Equation (10), all components are either hyperparameters or labels, with the exception of the function $k(\cdot, \cdot)$, traditionally a classical step. In the literature, various classical functions have been chosen for $k$, but our proposal is to use a quantum kernel from gate-based quantum computation.

Throughout our experiments, we utilized the Advance 4.1 machine from D-Wave [19] for all annealing experiments (more detail on the architecture can be found in Appendix B).

*2.2. From Classical to Quantum Kernel*

As previously discussed, the Gaussian kernel is the most used radial basis function (RBF) kernel. It facilitates the computation of the similarity between each pair of points in a higher-dimensional space while circumventing explicit calculations. Once an RBF kernel is selected, the similarities between each pair of points are computed and stored in a kernel matrix. Due to the symmetry of the distance function, it naturally follows that both the kernel matrix and, subsequently, $\tilde{Q}$ are symmetric as well.

In our proposal, we assume that the kernel is a Quantum Kernel. This approach allows us to leverage the exponentially higher dimension of the Hilbert space to separate the data effectively.

For the sake of simplicity, in the experiment we propose as a proof of concept, we employ the simple algorithm of the quantum inner product to compute the similarity between two vectors in the Hilbert space (Figure 2). Consider two points $x_i$ and $x_j$ along with their respective feature maps $A$ and $B$, such that $|x_i\rangle = A|0\rangle$ and $|x_j\rangle = B|0\rangle$ on a $M$ qubit register. Our objective is to derive the similarity $s$ between the two through the quantum inner product:

$$B^{\dagger}A|0\rangle = a_0|0\rangle + \sum_{k=1}^{2^M-1} a_k|k\rangle \tag{11}$$

$$a_0 = \langle 0|B^{\dagger}A|0\rangle = \langle x_j|x_i\rangle = s \tag{12}$$

where $|k\rangle$ and $a_k$ are, respectively, the $k$th standard basis vector and its coefficient. It is evident that if $x_i = x_j$, then $a_0$ is equal to 1, while if they are orthogonal to each other, the result will be 0. This behavior describes a similarity metric in a high-dimensional space, precisely what we sought to integrate into our Quantum Support Vector Machine.

**Figure 2.** Generic representation of the quantum inner product of two vectors in a $M$ qubit circuit. $A$ and $B$ represent feature map circuits that encode the two vectors, while $a_0$ is the similarity metric that we are looking for.

The final step to complete the QgSVM involves determining two feature maps, $A$ and $B$, capable of encoding the data in the Hilbert space. To obtain the quantum inner product between vectors, we employ the same feature map technique for both $A$ and $B$. Additionally, given that each vector $x_i$ for all $i \in \{0, \ldots, N-1\}$ possesses the same number of features $\gamma$, it follows that the number of qubits in every circuit used for gate-based computation depends solely on $\gamma$ and the chosen feature map technique.

In our study, we opt for one of the widely employed feature map techniques known as the Pauli expansion circuit [17]. The Pauli expansion circuit, denoted as $U$, serves as a data encoding circuit that transforms input data $\vec{x} \in \mathbb{R}^N$, where $N$ represents the feature dimension, as

$$U_{\Phi(\vec{x})} = \exp\left( i \sum_{S \in \mathcal{I}} \phi_S(\vec{x}) \prod_{i \in S} P_i \right). \tag{13}$$

Here, $S$ represents a set of qubit indices describing the connections in the feature map, $\mathcal{I}$ is a set containing all these index sets, and $P_i \in \{I, X, Y, Z\}$. The default data mapping is

$$\phi_S(\vec{x}) = \begin{cases} x_i \text{ if } S = \{i\} \\ \prod_{j \in S} (\pi - x_j) \text{ if } |S| > 1 \end{cases}. \tag{14}$$

This technique offers various degrees of freedom, including the choice of the Pauli gate, the number of circuit repetitions, and the type of entanglement. In our implementation, we select a second-order Pauli-Z evolution circuit, a well-known instance of the Pauli expansion circuit readily available in libraries such as Qiskit [20]. A more detailed formulation will be provided in the next subsection.

Our decision to use this specific feature map aligns with prior research by Havlicek et al. [17], where they leverage a Pauli Z expansion circuit to train a Quantum gate-based Support Vector Machine (QgSVM). It is essential to note that this choice is arbitrary and the feature map can be substituted with a variety of circuits. One important constraint applied to the feature map circuits is that, in a real-world implementation, they must encode data in a manner not easily simulated by classical computers. This requirement is necessary, though not sufficient, for achieving a quantum advantage.

Given that the focus of this paper is on proposing a new methodology rather than experimentally proving quantum advantage, we opted to simulate the quantum gate-based environment on classical hardware during the experimental phase.

### 2.3. Feature Map

As previously discussed, the feature map we chose for our experiments is a variation of the Pauli expansion gate known as the ZZ feature map. We employ a linearly entangled, single repetition, $N$-qubit ZZ feature map, illustrated in Figure 3. The rotation gates in this context involve rotations around the Z-axis:

$$R_Z(\theta) = \begin{pmatrix} e^{-i\theta} & 0 \\ 0 & e^{i\theta} \end{pmatrix} \tag{15}$$

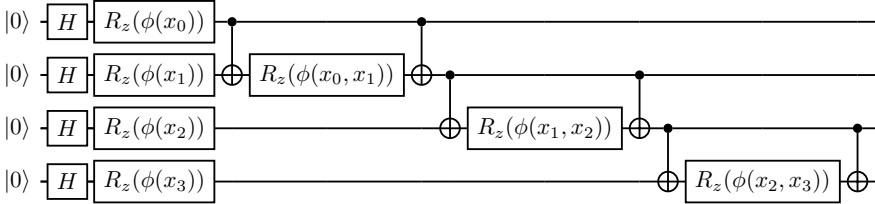

**Figure 3.** As an example of the feature map used, consider the circuit that takes input $\vec{x} = \{x_0, x_1, x_2, x_3\}$ and performs the encoding of classical data in the quantum space. The circuit depicted in the figure represents the first part of the inner product circuit. The second part involves the adjoint of this circuit with a different vector $\vec{z}$ as input.

The function $\phi(\cdot)$ is the one described in Equation (14) that assumes the default forms depending on how many parameters are passed:

$$\begin{aligned} \phi(x) &= x \\ \phi(x_0, x_1) &= (\pi - x_0)(\pi - x_1) \end{aligned} \tag{16}$$

Once the kernel is constructed, it can generates the similarity score between any pair of data points. We utilize this kernel to compile a kernel matrix $K$, where $K_{n,m} = k(x_n, x_m)$, similar to the classical SVM approach. As mentioned earlier, the matrix is symmetric over the main diagonal since $k(x_n, x_m) = k(x_m, x_n)$.

It is crucial to highlight the computational bottleneck in this methodology: obtaining the similarity score involves repeated executions of the circuit to derive the probability. Once we have the probability $p_0$ of the state $|0\rangle^{\otimes n}$, we can approximate $s$ to a value $\hat{s}$, which, for our purposes, serves as a valid proxy for $s$. Our proposal addresses this problem by generating an ensemble of models that compute smaller $K$. This approach is a standard practice in classical SVM, as documented in [21]. It has been demonstrated to enhance generalization and decrease the computational cost in the kernel part of the algorithm.

### 2.4. Experiments

Our experiments serve as a demonstration of the viability of the proposed methodology. All experiments involve binary classification problems conducted on pairs of classes from the MNIST dataset [22], following preprocessing. Due to the limited capabilities of current quantum computers and simulations, and to maintain simplicity for demonstration purposes, we applied Principal Component Analysis (PCA) [23] to each image, considering only the first two principal components as input to the model (Table 1 reports the explained variance). The resulting values are then mapped into the range $x_i \in [0, \pi/2]$ to fully exploit the properties of the Quantum gate-based Support Vector Machine (QgSVM). Simultaneously, the labels are adapted to the Quantum Annealer-based Support Vector Machine (QaSVM) standard, where $y_i \in \{-1, 1\}$.

**Table 1.** The sum of the explained variance based on the first two components from the PCA of each dataset. Principal Component Analysis generates new features to represent the data and orders them based on the variance they explain. Our implementation considers only the first two components.

| Dataset | Explained Variance |
|---------|--------------------|
| MN09 | 30.8% |
| MN38 | 21.0% |
| MN47 | 23.1% |
| MN56 | 25.4% |

We present four different experiments utilizing class pairs 0–9, 3–8, 4–7 and 5–6; referred to as MN09, MN38, MN47, and MN56, respectively. In each experiment, we selected 500 data points with balanced classes and divided them into a training set (300 data points) and a test set (200 data points), each with two features (referred to as $\gamma$).

The experiments are divided into three phases: hyperparameter tuning, training, and testing.

*Hyperparameter Tuning:* Initially, a 4-fold Monte Carlo cross-validation is performed on the training set for hyperparameter tuning. The optimized hyperparameters include $K$, $B$, and $\xi$ from Equation (8). While an exhaustive search on the hyperparameter space is beyond the scope of this work, additional information and results on the effect of hyperparameter tuning can be found in [9]. To assess the performance of each model, we compute the Area Under the Receiver Operating Characteristic curve (AUROC) and the Area Under the Precision–Recall curve (AUPCR), as shown in Figure 4.

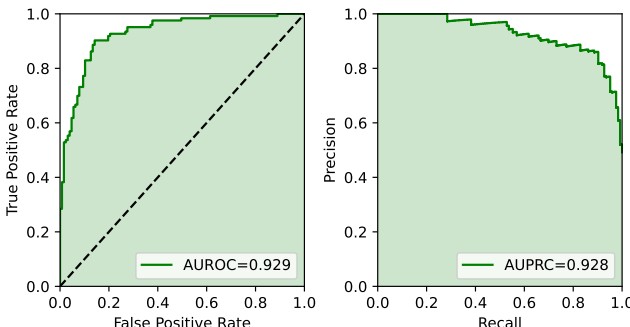

**Figure 4.** Example of AUROC and AUPRC curves. This graph illustrates the True Positive vs. False Positive curve (AUROC on the **left**) and the area under the Precision vs. Recall curve (AUPRC on the **right**) of a classifier when predicting classes for the training set on the dataset MN09 while performing hyperparameter tuning.

*Training:* Once the optimal hyperparameters are determined, we instantiate the best model for each dataset and train it on the entire training set. Following the original proposal by Willsch et al. [9], and to address the challenge of limited connectivity in the annealing hardware, the entire training set is divided into six small, disjoint subsets (referred to as folds), each containing approximately 50 samples. The approach involves constructing an ensemble of quantum weak Support Vector Machines, with each classifier trained on one of the subsets.

This process unfolds in two steps. First, for each fold, the top twenty solutions (qSVM($B, K, \xi$)#$i$ where $i \in \{1, \cdots, 20\}$ is the index of the best solutions) obtained from the annealer are combined by averaging their respective decision functions. Since the decision function is linear in the coefficients, and the bias for each fold $l$, denoted as $b^{(l,i)}$, is computed from $\alpha_n^{(l,i)}$ using Equation (4), this step effectively produces a single classifier with an effective set of coefficients given by

$$\alpha_n^{(l)} = \sum_i \alpha_n^{(l,i)} / 20$$

and bias given by

$$b^l = \sum_i b^{(l,i)} / 20$$

Second, an average is taken over the six subsets. It is important to note that the data points $(\mathbf{x}_n, y_n)^l$ are unique for each $l$. The complete decision function is expressed as

$$F(\mathbf{x}) = \frac{1}{L} \sum_{nl} \alpha_n^{(l)} y_n^{(l)} k\left(\mathbf{x}_n^{(l)}, \mathbf{x}\right) + b \tag{17}$$

where $L$ is the number of folds and $b = \sum_l b^{(l)} / L$. Similar to the formulation showed before, the decision for the class label of a point $\mathbf{x}$ is determined by $\tilde{t} = \text{sign}(F(\mathbf{x}))$.

*Testing:* After the training concludes, we test the models on the 200 data points of the test set. Similar to the training phase, a total of 6 weak SVMs, each resulting from the combination of the 20 annealer's solutions, are utilized to determine the class of the data point.

A visual representation of the process can be found in Figure 5, while in Appendix C a toy example with the visualization of the kernel and QUBO matrix can be found.

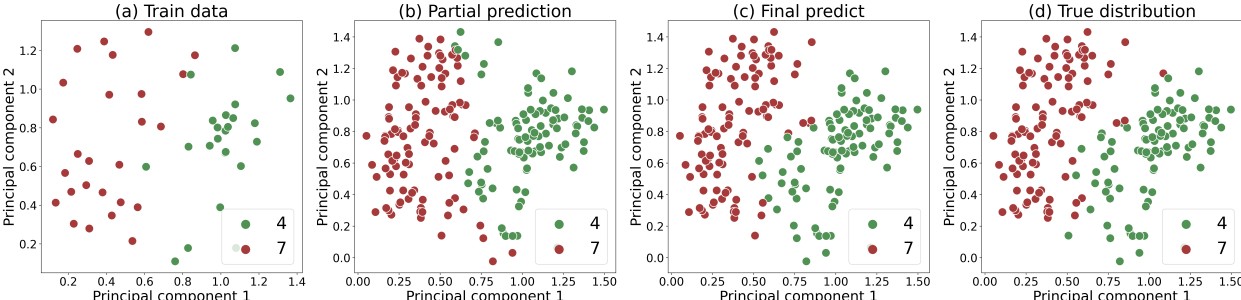

**Figure 5.** This image illustrates various stages of prediction. In (**a**), one of the subsets of the training set is displayed; a weak QSVM trains on it. (**b**) shows the prediction of the weak model on the test data, obtained by combining the 20 results computed by the annealer. (**c**) contains the result of aggregating the predictions from the six weak QSVMs. Lastly, in (**d**), the real distribution of labels on the test set is depicted. It can be observed that the partial prediction is significantly off-target with respect to the true distribution, while the final prediction is very close to it. This is partly the result of the folding of the training set, which allows the weak classifiers to see only a small subset of the training data, and partly stems from the intrinsic randomness inherent in quantum processes like quantum annealing.

## 3. Results

In this section, we present the results obtained from the new methodology. As discussed earlier, these experiments should be viewed as a proof of concept and as evidence of the correctness of the methodology, rather than an exhaustive search for the best possible model. We searched over a narrow hyperparameter space to obtain a model slightly superior to the most naive approach. For each dataset, we performed hyperparameters tuning in the space $B = \{2, 10\}$, $K = \{2, 3\}$, and $\xi = \{1, 2\}$, where B is the basis used to represent $\alpha$ (Equation (7)), K is the number of $a_k$ used to represent $\alpha$ (Equation (7)), and $\xi$ is the coefficient used when adding the constraint in Equation (8). The results of the tuning phase are reported in Appendix A, while the best hyperparameter for each model is shown in Table 2.

It is noteworthy to emphasize that the choice of hyperparameters can significantly impact the results, even with a relatively shallow search as the one we performed. The effect is particularly evident for the dataset *MN38* in Table A2, where the standard deviation between the data of every validation column is close to 0.1, while the difference between the maximum and minimum scores is more than 0.22 points in each validation column.

Once we have selected the best hyperparameters for each model, we proceed training six weak classifiers, each composed of 20 solutions returned by the quantum annealer. The combination of these six can be regarded as one model trained on the entire dataset. Finally, the model is tested on the test set. The results of each model are reported in Table 3, while Figure 6 illustrates the predictions on the test set for each dataset, highlighting the errors.

**Table 2.** Best results for hyperparameter tuning on each dataset. The columns B, K, and $\xi$ represent the hyperparameter values that obtained the best AUPCR score. T. (*Train*) and V. (*Validation*) in the column names stand for the training and validation sets. acc, AUROC, and AUPCR are accuracy, the area under ROC, and the area under PCR scores.

| Dataset | B | X | $\xi$ | T. acc | T. AUROC | T. AUPCR | V. acc | V. AUROC | V. AUPCR |
|---------|---|---|-------|--------|----------|----------|--------|----------|----------|
| MN09 | 2 | 3 | 1 | 0.9354 | 0.9839 | 0.9845 | 0.9375 | 0.9877 | 0.9886 |
| MN38 | 2 | 3 | 1 | 0.8667 | 0.9083 | 0.8391 | 0.8722 | 0.9372 | 0.9067 |
| MN47 | 2 | 3 | 1 | 0.9354 | 0.9742 | 0.9795 | 0.9014 | 0.9674 | 0.9708 |
| MN56 | 2 | 2 | 2 | 0.8937 | 0.9675 | 0.9640 | 0.8972 | 0.9648 | 0.9602 |

**Table 3.** This table shows the macro average of Precision, Recall, and f1-score for each dataset on the test set.

| Dataset | Precision | Recall | f1-Score |
| --- | --- | --- | --- |
| MN09 | 0.91 | 0.91 | 0.90 |
| MN38 | 0.84 | 0.80 | 0.80 |
| MN47 | 0.97 | 0.97 | 0.97 |
| MN56 | 0.89 | 0.87 | 0.87 |

The results are consistently positive, showcasing the models' high accuracy in predicting the classes of the datasets. Some errors can be attributed to the limited information contained in only two features, even though these features are the most expressive in Principal Component Analysis.

As hypothesized during formulation, the experiments provide evidence of the effectiveness of this approach and highlight the synergistic power of the two quantum technologies when employed together.

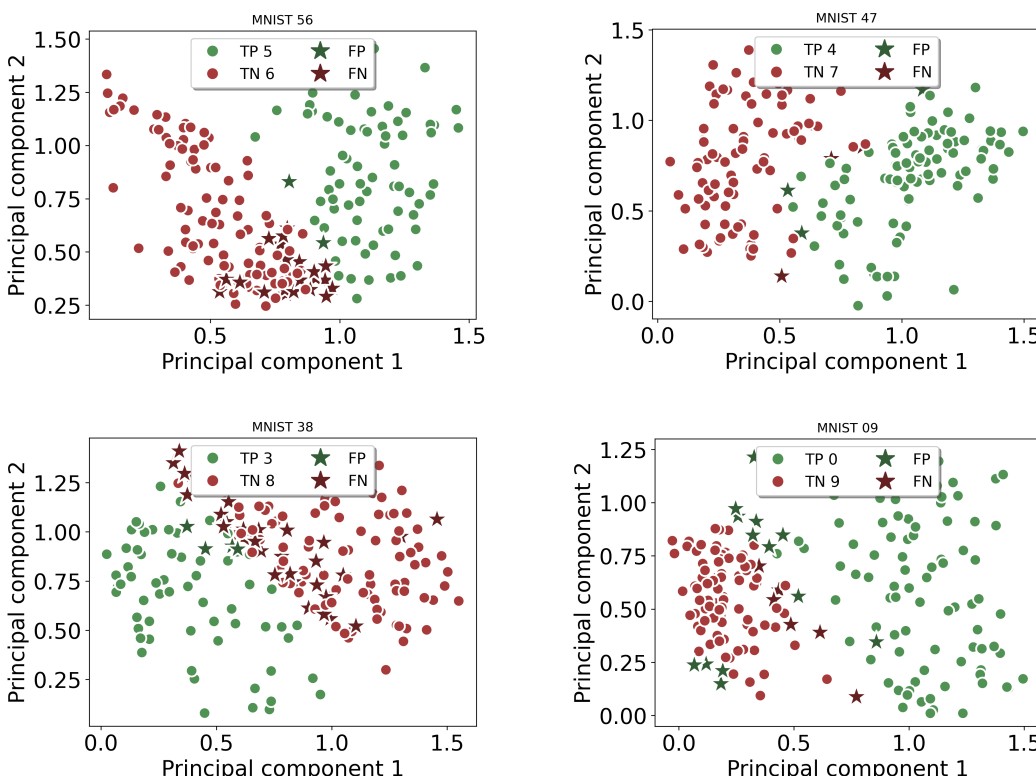

**Figure 6.** The figures illustrate the models' predictions on each dataset. The axes represent the first two component obtained with PCA and used for classification. Circles denote correctly classified instances, while stars indicate misclassifications. The color of the stars represents the class to which they have been erroneously assigned. Each graph considers the lower-numbered class as "Positive" and the higher-numbered class as "Negative".

## 4. Discussion

The results obtained from our Hybrid Quantum Support Vector Machine (Hybrid QSVM) showcase great promise across various aspects. Remarkably, we achieved favorable outcomes using only two features derived through Principal Component Analysis (PCA) while harnessing both quantum gate-based technology (simulated) and quantum annealing (real hardware).

As previously mentioned, our experimental focus was not centered on seeking the optimal model or comparing its performance against classical approaches. Instead, our primary goal was to experimentally validate the viability of our proposed approach. We deliberately invested minimal effort in performance optimization, aside from a modest hyperparameter tuning step that surpassed the naive approach. Consequently, these results serve as a baseline for potential future enhancements to the model.

During the tuning phase, we concentrated solely on hyperparameters related to the annealing component of the algorithm (specifically, $B$, $K$, and $\xi$), without delving into the gate-based part. This decision is motivated by three main factors.

Firstly, the existing literature on Quantum gate-based Support Vector Machines (QgSVM) is more developed compared to Quantum annealing Support Vector Machines (QaSVM), making exploration of the latter more pertinent from a research standpoint.

Secondly, we had access to real annealing hardware through the D-Wave Leap program [24]. Although access to real gate-based hardware is possible through the IBM Quantum Initiative, leveraging both technologies concurrently was impractical due to extended wait times in the hardware queues. Moreover, given our constraint of limiting each data point to two features, a two-qubit gate-based quantum hardware was sufficient for simulating the quantum kernel, further simplifying the process.

Third, and directly linked to the point above, we considered that an easy-to-simulate quantum kernel cannot surpass classical performance. This means that one of the reasons why the QgSVM in our experiment performed sub-optimally is because we could not access hard-to-simulate quantum hardware. For this reason, optimizing its hyperparameters was deemed futile. The same applies to the circuit design, we decided to keep it fixed for the whole experimental phase, since any 2-qubit circuit (with depth within reason) can be simulated.

It is essential to clarify that the three reasons outlined above pertain to the experimental setup and not the specific model's performance on the datasets. While tuning different hyperparameters or exploring more values of the already optimized ones might enhance performance on specific datasets, it deviates from the primary objective of this paper.

When talking about the goal of this proposal and the advantages it brings, we need to highlight that the quantum annealer is on the verge of addressing industrial optimization challenges [24]. Simultaneously, the inherent variational nature of the QgSVM positions it favorably for the Noisy Intermediate-scale Quantum (NISQ) era. The convergence of these two approaches is nearing completion, offering several potential benefits. QgSVM can leverage the exponentially large Hilbert space $\mathcal{H}$ to more effectively capture similarities and distances between points than classical kernel tricks. This allows it to distinguish between points that are traditionally considered challenging. Moreover, the quantum annealer's speed surpasses classical optimization methods, providing a solution quicker.

Recent studies on Quantum gate-based Support Vector Machines (QgSVM) have highlighted certain theoretical limitations under specific assumptions. In [25], the authors connect the exponential dimension of the feature space to a limitation in the model's ability to generalize. One proposed solution, discussed both theoretically [26] and empirically [27], involves introducing a new hyperparameter to regulate the bandwidth.

We argue that our proposal can further enhance the generalization ability of QgSVM. Building on the findings from the original QaSVM study [9], we know that Quantum annealing Support Vector Machines can outperform the single classifier obtained by classical SVM optimization in the same computational problem, and this can be directly translated to an improvement on the QgSVM since its optimization is done classically. This improvement stems from the ensembling technique used and the intrinsic ability of the annealer to generate a distribution of solutions close to optimal, thereby enhancing generalization (similar results can be found in [28,29] for different annealing machine learning models). Moreover, since our proposal is not tied to a specific implementation, it can employ various strategies to mitigate the limitations of QgSVM like the one discussed above.

Future work in this direction could involve a theoretical analysis of these features.

## 5. Conclusions

In the realm of quantum computing, the conventional approach involves selecting one technology and working within its framework, often neglecting the potential synergies that could arise from mixing different quantum technologies to harness their respective strengths. In this paper, we propose a novel version of Quantum Support Vector Machines (QSVM) that capitalizes on the advantages of both gate-based quantum computation and quantum annealing.

Our proposal positions itself within the context of Quantum Support Vector Machines (QSVM) from both technological points of view improving on both techniques. We enhance the generalization ability of standard QgSVM through the ensemble technique and leverage the intrinsic capability of Quantum annealing Support Vector Machines (QaSVM) to generate a distribution of suboptimal solutions during optimization within constant annealing time.

The ensemble method, similarly to its classical counterpart, not only enhances generalization but also reduces the computation time of the kernel matrix and the accesses to the quantum computer. For a dataset $X$ with $n$ samples and $\gamma$ features, the full kernel matrix is composed of $n^2/2$ elements, implying $\mathcal{O}(sn^2)$ calls to the (gate-based) quantum computer where $s$ is the number of shots needed to accurately compute the similarity score. Among other important factors [30], $s$ is strongly linked to the dimensionality of the Hilbert space [26] that in turn depends on the kernel implementation and $\gamma$. Our ensemble of $m$ Kernels quadratically reduces the needed number of accesses to $\mathcal{O}(s(n/m)^2)$ while only needing $\mathcal{O}(m)$ accesses to the quantum annealer. In this context, $m$ becomes a new hyperparameter and its value is highly dependant on the specific problem and the implementation in question.

To conclude, we can summarize our contribution in three points.

First, we introduce a method that combines quantum technologies, leveraging the complementarity between gate-based quantum computation and quantum annealing in the context of Support Vector Machines. While previous research has touched on the integration of these approaches for solving large-size Ising problems [31], to the best of our knowledge, our work is the first to explore this fusion in the context of classification problems.

Second, we provide experimental validation of our approach. Notably, the annealing component of our experiments is conducted on real quantum hardware, demonstrating the feasibility of our methodology in the early era of Noisy Intermediate-Scale Quantum (NISQ) computation.

Third, we establish a baseline result for future hybrid technology approaches, particularly on one of the most widely used datasets in classical machine learning.

Future work in this area is straightforward. As quantum technologies advance, our approach should undergo testing on quantum hardware to validate its capabilities. Additionally, there is a need for more extensive hyperparameter tuning to achieve optimal results. We recommend exploring and proposing different quantum kernel methods based on the characteristics of the data to further enhance performance. Simultaneously, a rigorous proof of the generalization power of our model is essential to prove the positive interaction of the technologies in a more rigorous manner.

It is crucial to note that the implementation of our methodology is not rigidly tied to the specific formulations described here. Rather, it adapts to the state of the art for each technology. The core of our proposal lies in the fusion of approaches, allowing for flexibility as advancements occur in individual technologies.

Quantum machine learning (QML) currently stands as a focal point of research, offering the potential to revolutionize various applications through the utilization of quantum computational power and innovative algorithmic models like Variational Algorithms. Similar to any scientific field, as it expands, new limitations come to light. However, simultaneously, researchers develop techniques to overcome and mitigate these restrictions. Despite the increasing attention, the field is still in its early stages, necessitating further exploration to unveil its practical benefits.

**Author Contributions:** Conceptualization, F.O.; Formal analysis, F.O. and S.G.; Investigation, F.O. and S.G.; Methodology, F.O.; Software, F.O. and S.G.; Supervision, S.L. and C.S.; Validation, F.O.; Writing—original draft, F.O.; Writing—review and editing, F.O. All authors have read and agreed to the published version of the manuscript.

**Funding:** This research received no external funding.

**Institutional Review Board Statement:** Not applicable.

**Informed Consent Statement:** Not applicable.

**Data Availability Statement:** All the data are publicly available. The dataset used for this study is the well-known MNIST. The code needed to reproduce the experiment is contained in the pubblic repository at https://github.com/filorazi/Hybrid_Quantum_Technologies_For_Quantum_Support _Vector_Machines/, (accessed on 30 November 2023).

**Acknowledgments:** We want to acknowledge the great help received by the people at the Julich research center. In particular thanks to Dennis Willsch and Gabriele Cavallaro that allowed the use of their code for our experiments.

**Conflicts of Interest:** The authors declare no conflicts of interest.

## Abbreviations

The following abbreviations are used in this manuscript:

| | |
|---|---|
| QUBO | Quadratic Uncontraint Binary Optimization |
| SVM | Support Vector Machine |
| NISQ | Noisy Intermediate-Scale Quantum |
| QaSVM | Quantum annealing Support Vector Machine |
| QgSVM | Quatum gate-based Support Vector Machine |

## Appendix A. Hyperparameter Tuning

This appendix contains the hyperparameter tuning tables for all datasets Tables A1–A4. Each table contains all combination of the three hyperparameters and for each combination presents the results on six different metrics: Accuracy on training set, Area under ROC on training set, Area under PCR on training set, Accuracy on validation set, Area under ROC on validation set, and Area under PCR on validation set. When choosing the best hyperparameters, we consider the AUPCR metric on the validation set.

**Table A1.** Results of hyperparameter tuning for the MN09 dataset. The columns *B, K,* and *ξ* represent the hyperparameter values. *Train* and *Val* in the column names stand for training and validation sets. *acc*, *AUROC*, and *AUPCR* are accuracy, the area under ROC, and the area under PCR scores. The best combination of hyperparameter based on *Val AUPCR* is B = 2 K = 3 $\xi$ = 1.

| B | K | $\xi$ | Train acc | Train AUROC | Train AUPCR | Val acc | Val AUROC | Val AUPCR |
|---|---|---|---|---|---|---|---|---|
| 2 | 2 | 1 | 0.8958 | 0.9781 | 0.9800 | 0.9306 | 0.9850 | 0.9868 |
| 2 | 2 | 2 | 0.9312 | 0.9832 | 0.9848 | 0.9236 | 0.9864 | 0.9876 |
| **2** | **3** | **1** | **0.9354** | **0.9839** | **0.9845** | **0.9375** | **0.9877** | **0.9886** |
| 2 | 3 | 2 | 0.9167 | 0.9780 | 0.9765 | 0.9347 | 0.9826 | 0.9843 |
| 10 | 2 | 1 | 0.8708 | 0.9330 | 0.9247 | 0.8931 | 0.9489 | 0.9522 |
| 10 | 2 | 2 | 0.8937 | 0.9731 | 0.9716 | 0.9194 | 0.9741 | 0.9739 |
| 10 | 3 | 1 | 0.8104 | 0.8774 | 0.8689 | 0.7931 | 0.8518 | 0.8364 |
| 10 | 3 | 2 | 0.8271 | 0.9036 | 0.8829 | 0.7861 | 0.8857 | 0.8582 |

**Table A2.** Results of hyperparameter tuning for the MN38 dataset. The columns *B, K,* and $\xi$ represent the hyperparameter values. *Train* and *Val* in the column names stand for training and validation sets. *acc*, *AUROC*, and *AUPCR* are accuracy, the area under ROC, and the area under PCR scores. The best combination of hyperparameter based on *Val AUPCR* is B = 2 K = 3 $\xi$ = 1.

| B | K | $\xi$ | Train acc | Train AUROC | Train AUPCR | Val acc | Val AUROC | Val AUPCR |
|---|---|---|---|---|---|---|---|---|
| 2 | 2 | 1 | 0.8333 | 0.8980 | 0.8650 | 0.8500 | 0.9150 | 0.8672 |
| 2 | 2 | 2 | 0.8500 | 0.8660 | 0.8183 | 0.8667 | 0.8990 | 0.8710 |
| **2** | **3** | **1** | **0.8667** | **0.9083** | **0.8391** | **0.8722** | **0.9372** | **0.9067** |
| 2 | 3 | 2 | 0.8521 | 0.8886 | 0.8521 | 0.8764 | 0.9103 | 0.8964 |
| 10 | 2 | 1 | 0.6854 | 0.7696 | 0.7340 | 0.6750 | 0.7591 | 0.7271 |
| 10 | 2 | 2 | 0.6729 | 0.7522 | 0.6990 | 0.6917 | 0.7770 | 0.6844 |
| 10 | 3 | 1 | 0.6562 | 0.7105 | 0.7340 | 0.6500 | 0.7102 | 0.7472 |
| 10 | 3 | 2 | 0.7625 | 0.8211 | 0.7732 | 0.7847 | 0.8520 | 0.8350 |

**Table A3.** Results of hyperparameter tuning for the MN47 dataset. The columns *B, K,* and $\xi$ represent the hyperparameter values. *Train* and *Val* in the column names stand for training and validation sets. *acc*, *AUROC*, and *AUPCR* are accuracy, the area under ROC, and the area under PCR scores. The best combination of hyperparameter based on *Val AUPCR* is B = 2 K = 3 $\xi$ = 1.

| B | K | $\xi$ | Train acc | Train AUROC | Train AUPCR | Val acc | Val AUROC | Val AUPCR |
|---|---|---|---|---|---|---|---|---|
| 2 | 2 | 1 | 0.9146 | 0.9688 | 0.9727 | 0.8931 | 0.9547 | 0.9595 |
| 2 | 2 | 2 | 0.8792 | 0.9597 | 0.9646 | 0.8792 | 0.9612 | 0.9615 |
| **2** | **3** | **1** | **0.9354** | **0.9742** | **0.9795** | **0.9014** | **0.9674** | **0.9708** |
| 2 | 3 | 2 | 0.9250 | 0.9772 | 0.9790 | 0.9111 | 0.9684 | 0.9706 |
| 10 | 2 | 1 | 0.7833 | 0.9015 | 0.9030 | 0.8306 | 0.9227 | 0.9226 |
| 10 | 2 | 2 | 0.8542 | 0.9486 | 0.9526 | 0.8736 | 0.9428 | 0.9450 |
| 10 | 3 | 1 | 0.7500 | 0.8701 | 0.8732 | 0.7167 | 0.8249 | 0.8378 |
| 10 | 3 | 2 | 0.8292 | 0.9182 | 0.9264 | 0.7972 | 0.8994 | 0.9061 |

**Table A4.** Results of hyperparameter tuning for the MN56 dataset. The columns *B, K,* and $\xi$ represent the hyperparameter values. *Train* and *Val* in the column names stand for training and validation sets. *acc*, *AUROC*, and *AUPCR* are accuracy, the area under ROC, and the area under PCR scores. The best combination of hyperparameter based on *Val AUPCR* is B = 2 K = 2 $\xi$ = 2.

| B | K | $\xi$ | Train acc | Train AUROC | Train AUPCR | Val acc | Val AUROC | Val AUPCR |
|---|---|---|---|---|---|---|---|---|
| 2 | 2 | 1 | 0.8896 | 0.9613 | 0.9566 | 0.8903 | 0.9375 | 0.9264 |
| **2** | **2** | **2** | **0.8937** | **0.9675** | **0.9640** | **0.8972** | **0.9648** | **0.9602** |
| 2 | 3 | 1 | 0.9062 | 0.9630 | 0.9608 | 0.8944 | 0.9519 | 0.9437 |
| 2 | 3 | 2 | 0.9229 | 0.9679 | 0.9677 | 0.9028 | 0.9527 | 0.9429 |
| 10 | 2 | 1 | 0.8292 | 0.9021 | 0.9028 | 0.8194 | 0.8918 | 0.8975 |
| 10 | 2 | 2 | 0.8042 | 0.8930 | 0.9074 | 0.8028 | 0.8955 | 0.9043 |
| 10 | 3 | 1 | 0.8354 | 0.9321 | 0.9328 | 0.8250 | 0.9264 | 0.9338 |
| 10 | 3 | 2 | 0.8813 | 0.9366 | 0.9483 | 0.8722 | 0.9436 | 0.9457 |

**Appendix B. Quantum Hardware and Simulation Detail**

In our experiment, we made use of both technologies of gate-based and annealing quantum computers. This appendix provides some technical information of the hardware used for quantum annealing and the simulation for gate-based computation.

*Appendix B.1. Gate-Based Simulation*

The kernel method explained in Section 2.2 is applied in practice thanks to the quantum computing local simulation offered by the Pennylane library [32]. The simulation runs on a classical system thanks to the Pennylane simulation on what is referred to as *default_qubit*. As reported in the official documentation, this names indicates *"a simple state simulator of qubit-based quantum circuit architectures"* that acts as a "device" able to backpropagate derivatives.

Additionally, although our specific implementation does not incorporate a trainable Quantum Support Vector Machine (QSVM) kernel, we utilized the *jax* interface to enhance the ability to differentiate and backpropagate information during training. JAX [33] is a system for transforming numerical functions, and Pennylane allows specifying it as the interface for classical backpropagation, enabling JAX to operate through the QNode.

The coding part was carried out in python with the Pennylane library.

As discussed in Section 4, the decision to utilize a simulation instead of a real quantum device is motivated by various factors. From a practical standpoint, employing real hardware would have necessitated $n^2/2$ requests. Given the current limitations and capacities of available gate-based quantum computers, coupled with the understanding that our experimentation is primarily a proof of concept, we opted to simulate the Quantum gate-based Support Vector Machine (QgSVM) component of the algorithm.

*Appendix B.2. Annealing Hardware*

The Quantum annealing Support Vector Machine (QaSVM) part, as detailed in Section 2.1, was executed using the available resources at DWave [24]. The specific hardware utilized is a quantum annealing computer known as *Advantage* [19], the Advantage QPU contains at least 5000 qubits and 15 couplers per qubit for a total of least 35,000 couplers. Notably, application problems can be more efficiently mapped onto the Advantage QPUs compared to its predecessor, the D-Wave 2000Q, as measured by chain length.

The advantage system uses a quantum chip with a Pegasus topology [19].

The choice of utilizing real hardware for the QaSVM has already been discussed in Section 4. From a practical perspective, the use of real quantum hardware increases the research value of our experiment. Moreover this not only testifies the validity of the approach but also function as further proof of current power of quantum annealers.

The use of the real annealer does not allow us to embed the full problem but this is only partially a limitation since, as we have discussed in Section 5, the use of an ensemble of models reduces the computational costs of the QgSVM and improves enhances generalization. It is essential to highlight that due to the inherent properties of the annealer, we naturally leverage multiple possible solutions it provides. Even if we could embed the entire problem, the solution would still be an ensemble comprising the best solutions outputted by the quantum computer.

The python code utilized for QaSVM has been made available to us from the autors of [9] and it has undergone minor changes. The code interfaces with the real hardware thanks to the Ocean library from D-Wave [18].

**Appendix C. Toy Problem and Visualization**

To enhance the clarity of the procedure, we present a small toy problem with a graphical representation. We have selected a total of four images from the MNIST09 dataset, which serve as training set. In the experiments, we reduce the dimensionality of these points using PCA, retaining only the two most relevant principal components. After the PCA, we map the values to the range $[0, \pi/2]$ to better encode them in a quantum state.

Once the data are pre-processed, our quantum kernel (QgSVM) is utilized to compute a similarity score. This toy problem employs the encoding strategy explained in Section 2.3 and represented in Figure 3. Each pair of points is passed through the kernel, and the obtained similarity is stored in a classical kernel matrix. For this instance of the MNIST09 problem, a $4 \times 4$ kernel matrix is generated, showing a high similarity score for points belonging to the class *9*.

Upon computing the entire kernel matrix, we create the QUBO matrix and encode it in the annealer. The QaSVM process then computes the solutions and outputs them as the binary encoding of $\alpha$ using Equation (7). Figure A1 shows the selected points, the kernel matrix and the QUBO matrix. The $\alpha$ values are then used to compute the predicted classes of each point.

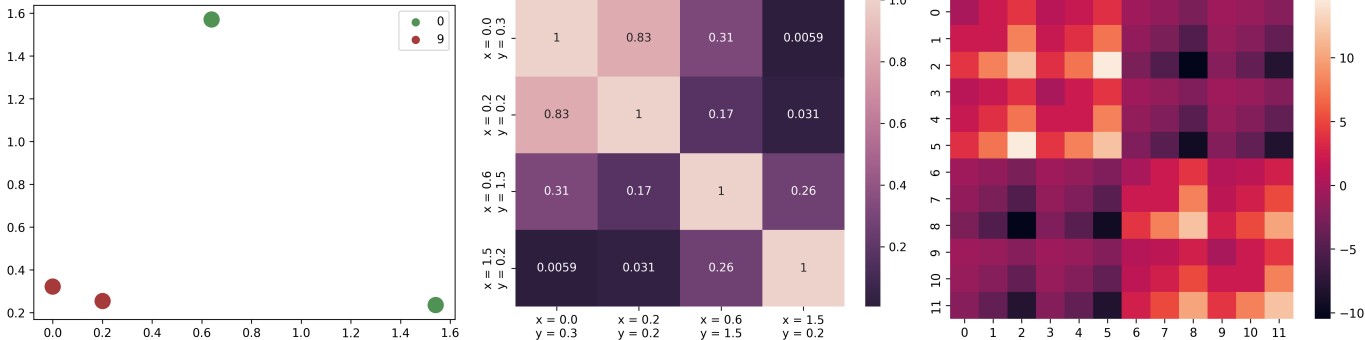

**Figure A1.** Graph representing the full transformation process for the toy problem. On the left is a scatterplot of the four points used in the toy problem, originating from MNIST09's images and reduced via Principal Component Analysis. In the center is the kernel matrix for the points of the toy problem with the selected quantum kernel. On the right is the QUBO matrix obtained from the kernel matrix.

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
