# Peer review of "Hybrid Quantum Technologies for Quantum Support Vector Machines"

_information, doi:10.3390/info15020072_

Round 1
Reviewer 1 Report
Comments and Suggestions for Authors
The manuscript by Orazi et al contains a nice idea which the authors execute: training/development of a quantum support vector machine (SVM) model can be carried out by calculating a kernel matrix on a gate-based quantum computer followed by finding the optimal SVM parameters (including support vectors) associated with that kernel matrix using a quantum annealer. This approach leverages the separate strengths of gate-based and annealer-based quantum computers. The authors make a compelling case, and the paper is relatively clearly written. However, there are a number of easily fixable but glaring issues with the manuscript in its current form as well as some larger issues with calculating kernel matrices in high dimensional Hilbert space which have been revealed in the quantum machine learning field which the authors need to address at least in their Discussion section.
Major comments:
1. The authors need to put their approach in better context with recent findings that calculating kernel functions in high dimensional Hilbert space is fraught with a number of challenges that can eliminate quantum advantage if proper conditions are not met. For example, Kubler et al find that quantum advantage depends on a quantum kernel's reproducible Kernel Hilbert space being low dimensional and functions are hard to compute classically (https://arxiv.org/abs/2106.03747). Canatar et al point out that generalization in quantum kernel models is hindered by the exponential size of the quantum feature space and show that under certain restrictions (i.e., changing the value of the bandwidth) can take a model from not being able to generalize to any target function to good generalization for well-aligned targets (https://openreview.net/pdf?id=A1N2qp4yAq). Thanasilp et al show that under certain conditions, values of quantum kernels over different input data can be exponentially concentrated (in the number of qubits) towards some fixed value which results in exponential scaling of the number of measurements required for successful training (https://assets.researchsquare.com/files/rs-2296310/v1/1f8b7fd87f353038f1ad06a0.pdf?c=1670551769). This is not an exhaustive list. The important point is that recent research in this area has pointed out severe challenges and issues with computing quantum kernels in high dimensional Hilbert space and the ability of SVM models using these kernels to generalize well if certain conditions are not met. How would these issues affect the authors approach and resulting models? They should at a minimum address this question in the Discussion section with appropriate references.
2. Figure 4 is not discussed or referenced anywhere in the manuscript. It should be references and briefly detailed in the Results section.
3. Figure 5 should be generated from one of the actual training/testing sets (e.g., MN38) and this information should be in the legend and described in the Results section.
4. Figures 2 & 3 are unreadable and are the result of a latex compilation error. All tables while readable also contain some latex rendering issue where extraneous text surrounds them.
5. Regarding Figure 6, the legend states that "the axes represent the most meaningful component obtained during PCA and used for classification". Which components were these and how much of the variance in the data did they each explain? This should be added to the Results section (and/or the Figure 6 legend) and the x and y axes should be labeled with these PCs. Moreover, on p. 7 lines 248-249, the authors write "Due to the limited capabilities of current quantum computers and simulations, we applied Principal Component Analysis (PCA) [22] to each image, considering only the first two components." Based on Figure 6 legend, the first two components were not necessarily the "most meaningful component" or were they? In fact, Figure 6 legend contradicts the text on p. 7 lines 248-249 if the first two PCs were not used for classification. This should be cleaned up/clarified.
6. The authors could choose to attempt their hybrid classification approach with many more features where they substitute the use of an actual D-wave quantum computer with a simulation of the Quantum Annealing part of building the QSVM model. This would go further in addressing the questions/issues posed in my comment 1 above. I believe that while the core idea in this manuscript is interesting, the questions/issues that challenge estimation and performance of high dimensional quantum kernels affect the proposed approach. If the authors can show that use of the quantum annealing step overcomes some (not all) of these, this paper will be of much greater interest to the community in my view. I leave this to the authors.
Minor comments
1. On p. 6, Eq. (12) can be simplified. Based on the authors definition of feature maps, |x_i> = A|0> and |x_j> = B|0>. Consequently, s = <x_j | x_i> = <0|B+A|0> = a_0 with the last equality coming from use of Eq. (11) and orthogonality of the states |k>. There is no need for the expression <x_j | B B+ |x_i> in Eq. (12).
2. On p. 6 line 210, the authors could site the Havlicek paper ([17] in current version of manuscript as follows: "...as the Pauli expansion circuit [17].".
3. On p. 8 line 275, the authors could add the definition of "L" as follows: "...where b=sum_l b(l)/L and L is the number of folds".
4. On p. 9 line 276, the authors should remove the "9" from "sign(F9(x))" so that it is "sign(F(x))".
Author Response
Dear Reviewer,
Thank you for your valuable insights and suggestions. We have carefully addressed each of the points you highlighted:
-
We conducted additional research to contextualize our work more effectively, and we incorporated comments in the manuscript to articulate how our research fits into the broader context.
-
Regarding Figure 4, we acknowledged the need for changes in the layout and have added relevant comments in the main text to provide further clarity.
-
We revised the caption for Figure 5 to explicitly state that it was generated during the experiments.
-
In response to the issues with Figures 2 and 3, we have replaced them with pdfs images to resolve potential LaTeX compilation errors. While we were unable to replicate the problem with the table, we made efforts to correct errors based on your description and feedback from other reviewers.
-
We have clarified the meaning of the features in the reduced dataset and added a table that presents the explained variance for each dataset to enhance understanding.
-
Regarding the impact of using more features on the annealing part, adding additional features would only affect the kernel computation and not the annealing process. Nevertheless, we added a discussion on the use of the ensemble method and how it contributes to our methodology.
We have addressed all minor comments and appreciate your thorough review. If you have any further suggestions or concerns, please feel free to let us know.
Reviewer 2 Report
Comments and Suggestions for Authors
I would like to begin this review by stating that, in my view, this paper has some strong points:
(1) In my personal opinion, the subject is very interesting, and their approach very ambitious.
(2) The organization of the paper is solid, the Figures and Tables are meaningful and interesting, and the whole presentation helps the reader to understand and appreciate the aim of the paper. This, combined with the references, can convince the reader that the authors know this area and have a good indication where their research should lead.
(3) The command of the English language is also very good, with very few typos (more on this later).
As someone that tries to use both annealers and IBM machines, I fully appreciate their ambitious approach. This is the reason that I would like to read more details about their implementations, say a schematic of the circuit, or the number of qubits used in D-Wave, etc.
Ideally, if it is possible, it would go a long way towards improving their exposition, if they could add one or two toy scale examples to illuminate the details.
I would also advise the authors to better and clearly explain the novelty and contribution of their work in the Introduction and extend Section 5.
Finally, the paper has very few typos and some strange formatting that could be addressed.
Page 1, line 29: “analisys”
Occasionally, in the text some equations are referenced a bit unconventionally, e.g., page 5, line 184: “equation 10” instead of “equation (10)”. This phenomenon occurs a couple of times in the text.
Figures 2 & 3 seem to have a formatting issue, i.e., “[column sep=2em]”, “[column sep=3pt, row sep=5pt]”. I was wondering if they are just formulas, or they also contain some missing pictures.
Tables 1 & 2 also seem to have a formatting issue, i.e., “2gray!5gray!25”, “2gray!5gray!25”.
The same pattern can be observed in the Tables A1-A3 in the Appendix A.
In conclusion, I find this paper very intriguing, well-written, and of interest to a very wide audience of the quantum community. I believe that the authors should do just a bit more writing to better explain their work so that this paper can be published.
Author Response
Dear Reviewer,
Thank you for your constructive feedback. We have taken the following actions to address your comments:
-
We are in the process of creating an appendix for the toy example, which will be included in the second round of review due to time constraints. In the meantime, we have revised the experiment description to enhance clarity.
-
The introduction has been refined, and we expanded Section 5 to provide a clearer explanation of the novelty of our approach, in line with your suggestions.
-
All minor comments, including formatting and grammar issues, have been attended to. Regarding the table, while we couldn't replicate the problem, we have made adjustments based on your description and feedback from other reviewers.
We appreciate your thorough review and are committed to addressing any further concerns or suggestions you may have.
Reviewer 3 Report
Comments and Suggestions for Authors
The paper is in general not easy to read with many definitions not properly defined and missing figures. For example, Fig2 and Fig3 were not compiled properly. I found even the first definition of SVM training loss in equation (1) not properly defined. Can't this be defined more compactly and in a more simpler form?
The main idea of combining quantum gate-based SVM and quantum annealer also needs more clarification, specifically regarding the implementation of quantum kernel and computing its value. See equations (11) and (12), what are a_k's here?
Comments on the Quality of English Language
The paper is written moderately well.
Author Response
Dear Reviewer,
Thank you for your valuable feedback. we addressed them as best as we could:
- We exchange all the latex text that could resolve in a formatting error (like figure 2 and 3) with pdf figures.
- Clarifications have been added to equations 11 and 12. ​
- Equation 1 presents the Quadratic Programming version for Support Vector Machines (SVMs), as it is widely used and can be translated into a QUBO formulation later on. However, we are open to deriving it if you believe it would enhance the manuscript.
We apologize for any inaccuracies in our definitions. To ensure that we address them appropriately, could you provide a more detailed list in the next round of revision?
Round 2
Reviewer 1 Report
Comments and Suggestions for Authors
The authors have addressed my comments.
Regarding the authors response to point 6, they make a valid point, but it's not obvious to me that as the feature space and hence Hilbert space dimension increases significantly, that this approach does not suffer from issues like Kernel concentration reported in Thanasilp et al. Specifically, a concentrated Kernel matrix could result in challenges with optimization (as detailed in Thanasilp et al) during the authors annealing step. Imagine supplying the quantum annealer with a random Kernel matrix! That would also challenge the annealing step and (hopefully) result in a poorly performing classifier. The authors don't need to address this point. They reference Thanasilp et al and offer a practical mitigation strategy via estimating an ensemble of m Kernels. However, it's worth noting that while this mitigation reduces the number of needed shots/measurements of the gate based quantum computer quadratically, the number of required shots/measurements increases exponentially with the size of the feature (hence Hilbert) space. Hence, the problem will remain for a large enough input feature space.
Minor comments
1. On p. 7 line 230, $I$ should be $\mathcal{I}$ (as it appears in Eq. (13)) or it will be confused with the identify operator shown as part of the Pauli set.
2. In Fig.. 4 legend, "...classifiers..." should be "...classifier...".
3. On p. 9 line 307, "...hyperparameter..." should be "...hyperparameters...".
4. On p. 12 line 378, a comma should be added between "problem" and "and" as follows: "...problem, and...".
5. On p. 12 line 385, "...feature." should be "...features".
6. On p. 12 line 401, "...sample..." should be "...samples...".
7. On p. 12 line 402, "...call..." should be "...calls...".
8. On p. 12 lines 406 and 407, "...access..." should be "...accesses...".
9. On p. 13 line 410: "...conlude..." and "...point:" should be "...conclude..." and "...points:", respectively.
Author Response
Thank you for the corrections; we have addressed all the minor comment you pointed out.
Regarding the impact of feature space size on generalization, we acknowledge that our ensemble method does not fully address the limitation discovered by Thanasilp et al. However, our belief is that considering fewer points not only results in a smaller kernel matrix but also diminishes the necessity for a high-dimensional feature space, as the fewer points may be easier to separate for the weak model. It's important to note that, at this point in time, this remains a speculative aspect requiring more thorough investigation before being explicitly stated in a research work. Consequently, we choose not to include it in the paper but rather work on this in the future to provide a solid proof.
Reviewer 2 Report
Comments and Suggestions for Authors
I am glad to see that all the issues I had raised have been addressed. Therefore, my final recommendation is to accept their work for publication.
Author Response
Thank you again for your suggestion and your help.